DATA RELEASE

# Kinship analysis and pedigree reconstruction by RAD sequencing in cattle

Yiming Xu[1],[†], Wanqiu Wang[2],[†], Jiefeng Huang[3],[†], Minjie Xu[4], Binhu Wang[2], Yingsong Wu[4], Yongzhong Xie[1] and Jianbo Jian[2],[5],[6],[*]

1 Animal Husbandry and Aquatic Affairs Center, Lianyuan City 417100, Hunan Province, China
2 BGI Genomics, BGI Center, 9 Yunhua Road, Yantian District, Shenzhen, 518081, China
3 Loudi Municipal Bureau of Agriculture and Rural Affairs, Loudi City 417000, Hunan Province, China
4 People's Government of Shexian County 056400, Hebei Province, China
5 Department of Biotechnology and Biomedicine, Technical University of Denmark, Lyngby, 2800, Denmark
6 Guangdong Provincial Key Laboratory of Marine Biotechnology, Shantou University, Shantou, 515063, China

## ABSTRACT

Kinship and pedigree, used for estimating inbreeding, heritability, selection, and gene flow, are useful for breeding and animal conservation. However, as the size of crossbred populations increases, inaccurate generation and parentage assignment in livestock farms increase. Restriction-site-associated DNA sequencing is a cost-effective platform for single nucleotide polymorphism (SNP) discovery and genotyping. Here, we performed a kinship analysis and pedigree reconstruction for Angus and Xiangxi yellow cattle. A total of 975 cattle, including 923 offspring with 24 known sires and 28 known dams, were sampled and subjected to SNP discovery and genotyping. The identified SNP panel included 7,305 SNPs capturing the maximum difference between paternal and maternal genome information, allowing us to distinguish F1 from F2 generations with 90% accuracy. In conclusion, we provided a low-cost and efficient SNP panel for kinship analyses and the improvement of local genetic resources, which are valuable for breed improvement, local resource utilization, and conservation.

**Subjects** Genetics and Genomics, Animal Genetics, Statistics and Probability

**Submitted:** 25 January 2024

\* Corresponding author. E-mail: jbjian@126.com

† Contributed equally.

Preprint submitted at https://doi.org/10.1101/2024.05.20.595066

## INTRODUCTION

The selection of superior individuals as parents in crossbreeding systems is a general tool for genetic improvement [1]. Genetic selection is a new approach to selective breeding based on high-density markers covering the entire genome; it shortens the breeding cycle and reduces breeding costs. Predicting progeny from parental lines and selecting the best crosses rely on the accuracy of the pedigree; however, parentage assignment is expensive and laborious in standard livestock production systems. Therefore, the development of reliable genetic markers for kinship analyses and pedigree reconstruction is necessary to meet industry demand and improve livestock management systems.

Single nucleotide polymorphisms (SNPs) are abundant, widespread in the genome, and follow simple models of evolution [2]. There are various genotyping approaches, including bovine SNP chip, restriction-site-associated DNA sequencing (RAD-seq) [3], multiplexed shotgun genotyping [4], exome sequencing [5], and whole-genome resequencing (WGS).

Traditional SNP genotyping methods, which necessitate specific probes or primers tailored to each SNP of interest, can incur substantial costs, particularly for large-scale genotyping efforts. Moreover, these processes are often labor-intensive and time-consuming, especially in the absence of high-throughput methodologies. Additionally, they may necessitate sophisticated bioinformatics for data analysis, further augmenting the associated costs. In contrast, RAD sequencing generally employs a limited set of restriction enzymes and barcodes, which can significantly reduce reagent costs compared to individual SNP assays. Despite the complexity of next-generation sequencing data, a plethora of tools and pipelines, some open-source, are available to manage RAD data, thereby mitigating the costs of data analysis. Consequently, RAD sequencing is adept at handling large-scale projects and proves more cost-effective for the genotyping of numerous samples or a comprehensive set of SNPs. RAD-seq's appeal extends beyond its economic and temporal advantages, as it is also adept at analyzing complex genomic sequences, unconstrained by the size of the genome. This technique has been effectively leveraged in the construction of linkage maps, comparative genomics, and population genetics studies across a diverse array of organisms.

Parentage analysis can be divided into two categories: exclusion and likelihood [6]. The former is designed to identify incompatibilities between pairs of individuals to prove that one cannot be the parent of the other [7]. Parentage assignment using likelihood is a well-developed approach [8, 9]. However, unlike the rapid development of parentage assignment methods and related software over the past decade [6], little work has focused on generation classification, which is an important first step.

Due to its high carcass yield, rapid growth rate, and marbled meat quality, the Angus breed has become the predominant cattle breed in numerous countries. Xiangxi yellow cattle is an indigenous Chinese breed and was included in the *National Protection List of Livestock and Poultry Genetic Resources of China* in 2006 [10]. This breed can feed on low-quality roughage, is good at climbing hills, and is well-adapted to high temperatures. To improve meat quality and quantity, Angus was crossbred with Xiangxi yellow cattle, creating a new hybrid known as Xiangzhong black cattle, which captures the beneficial traits of each breed. The F1 generation backcrossed with sires to produce the F2 generation. To reduce breeding costs, it is common to breed F1 and F2 generations in one barn after crossbreeding. However, as the size of the crossbred population increases, inaccurate recording of the generation and parentage are more frequent due to error. Therefore, assigning individuals to each generation of the crossbreeding program is necessary, especially for genetic improvement and local resource conservation.

In the F1 generation, the phenotype of the offspring is intermediate between those of the two parents, making it difficult to distinguish between the F1 and F2 generations by visual inspection alone. In the absence of accurate records from local livestock farms, it is nearly impossible to distinguish between the F1 and F2 generations. Therefore, in this study, we evaluated 975 cattle from livestock farms by RAD-seq, including sires (Angus), dams (Xiangxi yellow cattle), the F1 generation from a cross between dams and sires, and the F2 generation from the backcross between F1 and sires and the intercross between F1. Our aims were (1) to distinguish between the F1 and F2 generations according to Mendelian laws of inheritance and (2) to construct an SNP panel with high confidence for kinship analysis and pedigree reconstruction for the Xiangzhong black cattle.



## MATERIAL AND METHODS

### Sample information

The samples were collected from a local farm in Loudi, Hunan Province, China. In total, 975 cattle were sampled, including 923 calves, 28 dams, and 24 sires. First, 28 Xiangxi yellow cattle dams were crossed with 24 Aberdeen-angus sires. Semen was collected from Angus bulls and crossed with females of the F1 generation to produce the F2 generation. All F1 and F2 individuals were cultured together in livestock farms. Mating between bulls and cows of the F1 generation was possible. Sires and dams were clearly labeled; however, for offspring cattle, the paternal line was ambiguous.

### Library preparation and RAD-seq

For DNA extraction, fresh blood was collected and prepared according to the protocol of solution-based DNA extraction methods [11]. After quality control, DNA samples were subjected to single-digest RAD-seq as described previously, with slight modifications [3, 12]. Briefly, samples meeting quality standards were digested with TaqI restriction endonuclease for 20 min at 37 °C and then randomly fractionated by Covaris Focused-Ultrasonicator. Then, 5 μl of RAD adapter was added to the interruption product. The products were cyclized by rolling circle amplification. The libraries were sequenced on the BGISEQ-500 platform (RRID:SCR_017979) (BGI, Shenzhen, China) [13].

### SNP genotyping and selection

The raw sequence data were filtered and trimmed using SOAPnuke (RRID:SCR_015025) [14] and separated according to the unambiguous barcodes and the specific enzyme recognition site. Each RAD read was mapped to the *Bos taurus* (NCBI:txid9913) genome [15, 16] using BWA v0.7.12-r1039 (RRID:SCR_010910) [17, 18]. The bam format was sorted and indexed using Samtools v1.14 (RRID:SCR_002105) with default parameters except for markdup [18]. Then, the format was converted to SAM format using Picard v2.26.10 (RRID:SCR_006525) [19]. GATK v4.1.2 (RRID:SCR_001876) was used for SNP and insertion-deletions (InDels) calling with the Variant Call Format (VCF) file having default parameters [20].

Quality control of candidate SNPs was performed using GATK "VariantFiltration." All SNPs that met the criteria for minimizing false positives (QUAL < 250.0, DP < 1500, DP > 6,000, MQ < 50.0 ‖ QD < 4.0 ‖ FS > 15.0 ‖ BaseQRankSum < −4.0 ‖ ReadPosRankSum < −3.50 ‖ MQRankSum < −10.0, SOR > 4.0, AN < 975, MQ0 ≥ 30) were considered potential high-quality SNP markers for subsequent analyses.

For the classification of generations, considering that there were 24 and 28 sires and dams, respectively, alleles with counts of 48 and 56 indicated homozygous loci. To expand the criteria to obtain adequate loci, we allowed for five different alleles, which could reflect sequencing errors. Therefore, we selected allele counts of 43–48 and 51–56 in the paternal and maternal genomes, along with different allelic genotypes, for further analysis. Loci with allele depths below six were excluded from the SNP set.

For parentage assignment, SNPs with more than two alleles were first excluded. The retained SNPs from paternal data meeting the following criteria were extracted to form an allele set: minor allele frequency > 0.25, heterozygosity > 0.2, proportion of missing data < 0.7, *p*-value for Hardy–Weinberg equilibrium > 0.01. Maternal and offspring data were filtered using this allelic set. Finally, sites with an allele depth < 6 and a proportion of missing data > 0.8 were excluded.

## Classification of F1 and F2 generations

All samples were genotyped based on the SNP panels used for generation classification, and loci that were identical to the maternal/paternal allele or heterozygous were counted. The heterozygous ratio was defined as the number of heterozygous SNPs divided by the total SNPs genotyped. The number of identical maternal/paternal SNPs divided by the total number of SNPs successfully genotyped was defined as the ratio of SNPs inherited from maternal/paternal parents. Samples with a heterozygous ratio > 0.5 were classified as F1, and the remaining samples were assigned to the F2 generation.

Scatter diagrams of the results were generated using R. For phylogenic tree construction, the *p*-distance matrix was calculated using VCF2Dis-1 (RRID:SCR_022513) [21], and a neighbor-joining tree was generated using PHYLIPNEW with fneighbor (v3.69.650) [22]. Finally, visualization was performed using Mega [23].

The population genetic structure was evaluated using the program Admixture (RRID:SCR_001263). The number of assumed genetic clusters K was set to 5, with 10,000 iterations for each run. The SNP selection was performed based on the following strategies: SNPs were chosen from loci with frequencies ranging from 43 to 48 and 51 to 56 among paternal and maternal data, respectively, which exhibited distinct allelic genotypes. The SNP panel comprised a total of 7,305 loci. Identity-by-descent (IBD) was analyzed using the PLINK (RRID:SCR_001757) [24] kinship matrix generated based on 7,305 SNPs using the kinship matrix function (-method Centered_IBS) in TASSEL (SCR_012837) [25] (v5.0) and visualized using a heatmap in R using the library "pheatmap" library [26] (https://github.com/raivokolde/pheatmap).

## Parentage assignment

Parentage assignment was performed using CERVUS (v3.0.7; RRID:SCR_025446) with the paternity selection function based on the likelihood approach [9]. Based on a simulation, 959 loci showed 91% successful assignment at a 95% confidence level. The R package APIS (v1.0.1; RRID:SCR_025445) [27] was used for parentage assignment based on the distribution of Mendelian transmission probabilities. The assignment error rate was set to 0.05. The first candidate sire given by the software was considered the most probable sire for the offspring and was used for further analyses. Exclusion-based parentage assignment was performed using the hiphop package (v0.0.1) in R [28]. None of the dams and sires were defined as social parents. The first-ranked candidate parent pair was selected as the most likely parent and subjected to subsequent analyses.

## RESULTS

### Discovery of SNPs by RAD-seq

To enhance the quality of beef and maintain local genetic resources, local farms introduced Aberdeen Angus as sires and crossed them with a local breed, Xiangxi yellow cattle (Figure 1A). Female F1 cattle were backcrossed with sires to produce the F2 generation (Figure 1A). Theoretically, if a sire and dam have different genotypes (AA and aa), the F1 generation will be heterozygous (Aa), and the F2 generation will contain two genotypes (AA, Aa). The F2 generation can be used for further crossbreeding, genetic improvement, and resource conservation. To reduce costs, farmers usually do not breed F1 and F2 generations in separate barns. Furthermore, mating is possible between bulls and cows of the F1 generation. Angus cattle and Xiangzhong Yellow cattle differ mainly in body color and size,

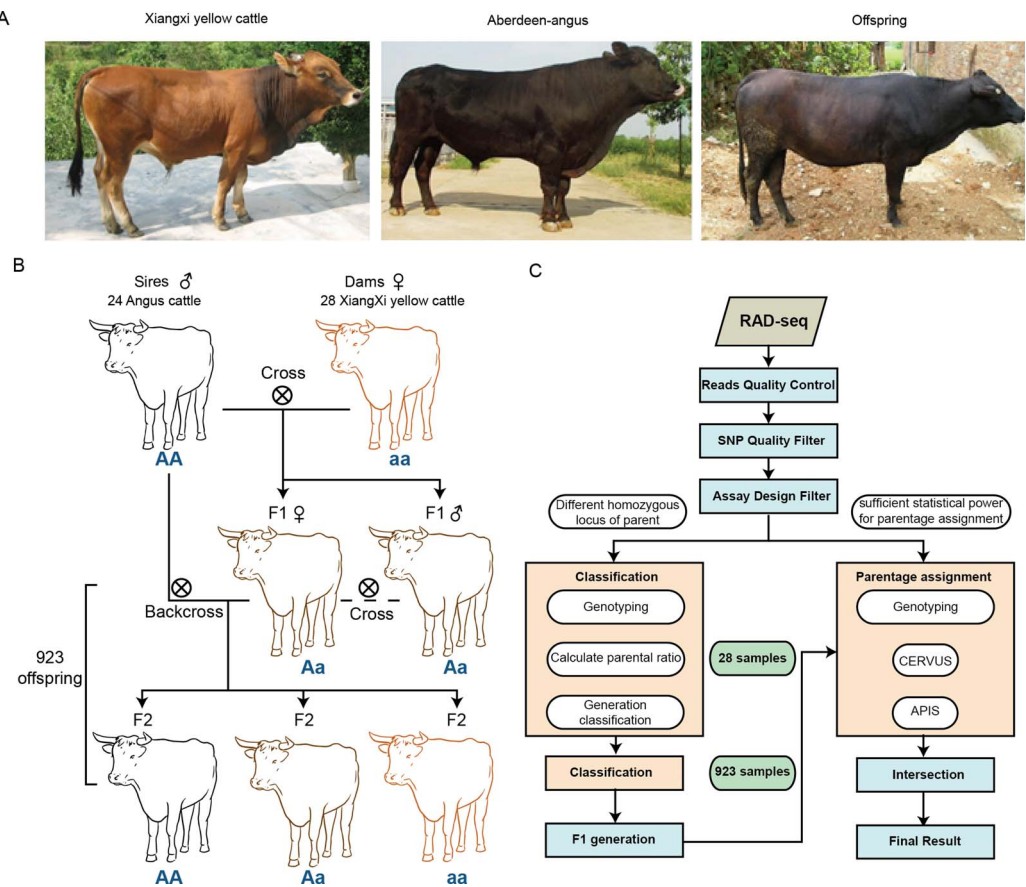

**Figure 1. Workflow for generation classification and the selection of a SNP panel retaining parental information.**
(A) Representative pictures of the three breeds. (B) Schematic diagram of the crossbreeding system. (C) Flowchart detailing the workflow for generation classification and parentage assignment.

i.e., Angus cattle are black and Xiangzhong Yellow cattle are brown. However, their offspring are mainly black; hence, it is difficult to distinguish between F2 progeny based on appearance without genotyping. The phenotype of F1 hybrids is similar to that of F2 carrying AA and Aa genotypes. In this study, we evaluated 975 cattle by RAD-seq, including 923 offspring with 24 known sires and 28 known dams. All samples were collected from Hunan province, China.

A total of 4.65 Tb of clean data were obtained from the samples, with a mean of 35.59 million reads, 1.59× depth, and 31.06% coverage per individual. An average of 35.96 million reads were mapped to the *B. taurus* reference genome ARS-UCD1.2, with an average mapping rate of 99.8%.

Genetic variants were initially detected by GATK and then underwent preliminary filtering. A total of 8,155,418 SNPs and 1,100,880 InDel variants that satisfied the criteria were retained (Tables 1 and 2 in GigaDB [29]). A density plot revealed that SNPs and InDels were preferentially distributed in the proximal telomeric regions (Figures 2 and 3). NC_037357.1, the X chromosome of *B. taurus*, had relatively few SNPs and InDels. The X chromosome is present in a single copy in males. Compared with autosome ones, the

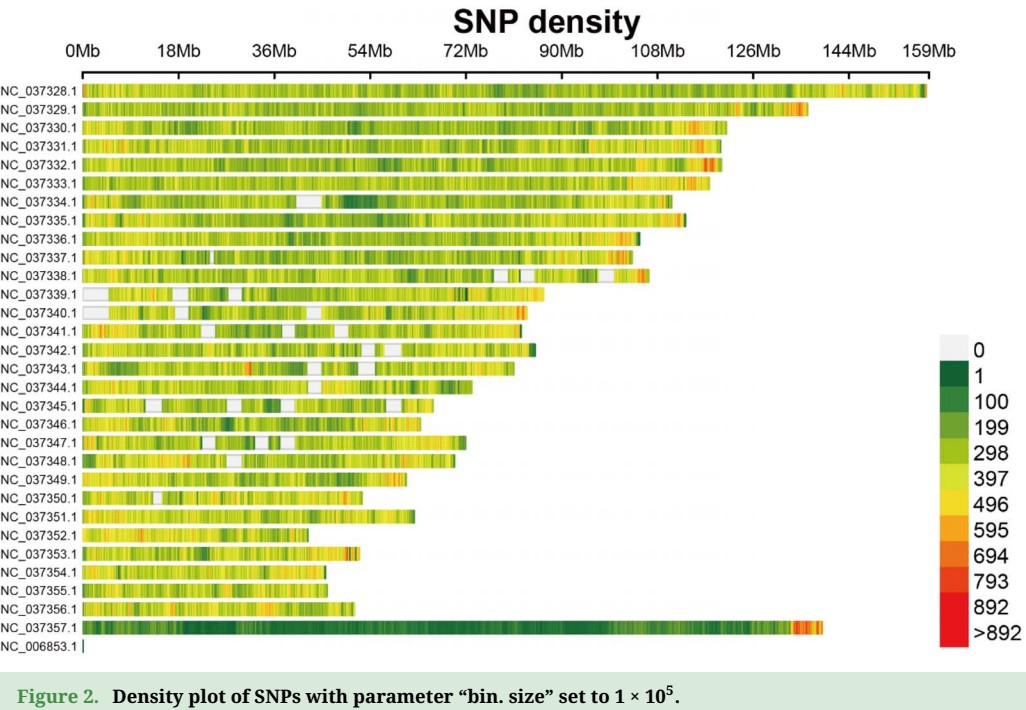

**Figure 2.** Density plot of SNPs with parameter "bin. size" set to $1 \times 10^5$.

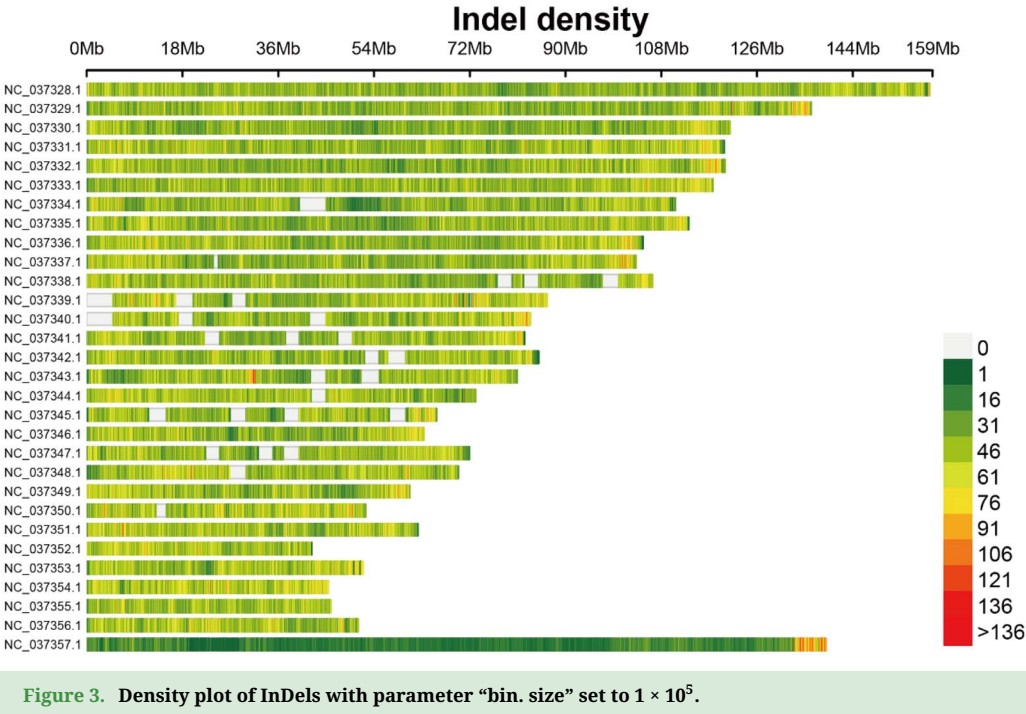

**Figure 3.** Density plot of InDels with parameter "bin. size" set to $1 \times 10^5$.

X chromosome has a lower recombination rate, lower mutation rate, and smaller effective population size [30].

## Construction of a SNP panel retaining parental information

Our first aim was to assign cattle lacking clear identifying information to each generation of the crossbreeding program. We had accurate generation information for 28 offspring from the farm record. According to Mendel's laws of inheritance, in a dominant-recessive inheritance system, parents with different homozygous genotypes (AA and aa) produce heterozygous F1 offspring (Aa). In the backcross of F1 to sires, all F2 progeny show the dominant trait: 50% are homozygous and 50% heterozygous, on average. If the F1 are intercrossed, it is expected that, on average, the F2 generation is 25% homozygous with the dominant trait, 50% is heterozygous showing the dominant trait (genetic carriers), and 25% is homozygous with the recessive trait. The genotypic ratio is 1 (AA):2 (Aa):1 (aa), and the phenotypic ratio is 3:1. Accordingly, homozygous loci from parents with different genotypes may provide pivotal information for identifying the generation of individuals (Figure 1B).

Retained SNPs were further filtered to ensure the maximum difference between paternal and maternal genome information. The workflow used to generate a high confidence SNP panel consisting of a series of homozygous loci is provided in Figure 1C. First, we chose SNPs that were homozygous in sires (AA) and dams (aa) for alternative alleles. However, only 45 loci were kept from 24 paternal and 28 maternal SNPs. Considering that the sequencing depth of some samples was not sufficient for genotyping, the criteria were expanded slightly to obtain adequate loci. The loci with frequencies of 43–48 and 51–56 among paternal and maternal data and exhibiting different allelic genotypes were selected as candidates for classification. Finally, the SNP panel included 7,305 loci.

Subsequently, we constructed a phylogenetic tree based on the panel with data for 24 sires, 28 dams, 15 offspring of the F1 generation, and 13 offspring of the F2 generation. All sires and dams were clearly assigned to distant groups in the phylogenetic tree. In addition, the F1 and F2 generations were generally separated (Figure 4A). However, it was still difficult to distinguish between F1 and F2 based on these data. Accordingly, it is necessary to consider both the percentage of heterozygous loci and the percentage of loci inherited from dams and sires.

## Assigning individuals to F1 and F2

From data for the F1 and F2 generations, we extracted the genotypes of all samples using the high-confidence SNP panel described above and compared them with the genotypes of sires and dams. The numbers of SNP loci in offspring with the maternal/paternal genotype (maternal /paternal genotype locus, MGL/PGL) were used to calculate the maternal/paternal ratio, and heterozygous loci (H-genotype) were used to calculate the heterozygous ratio. The maternal/paternal ratio was defined as the ratio of MGL/PGL to all genotypes. As mentioned above, all samples belonging to the F1 generation should have heterozygous alleles, and these alleles should segregate independently in the F2 generation. However, since we extended the criteria for the SNP panel, it is possible to obtain MGL or PGL in the F1 generation. Consequently, the offspring samples with heterozygous loci percentages > 0.5 were assigned to the F1 generation, and the rest were classified as F2. In the F2 generation, offspring with a maternal/paternal ratio > 0.5 were defined as having the most maternal/paternal characteristics.

As expected, 21 out of 28 offspring were correctly assigned based on farm records. A scatter diagram of all samples (Figure 4B) revealed four groups corresponding to sires (uppermost group), dams (lower), F1 (left corner), and F2 (near dams). Five samples were

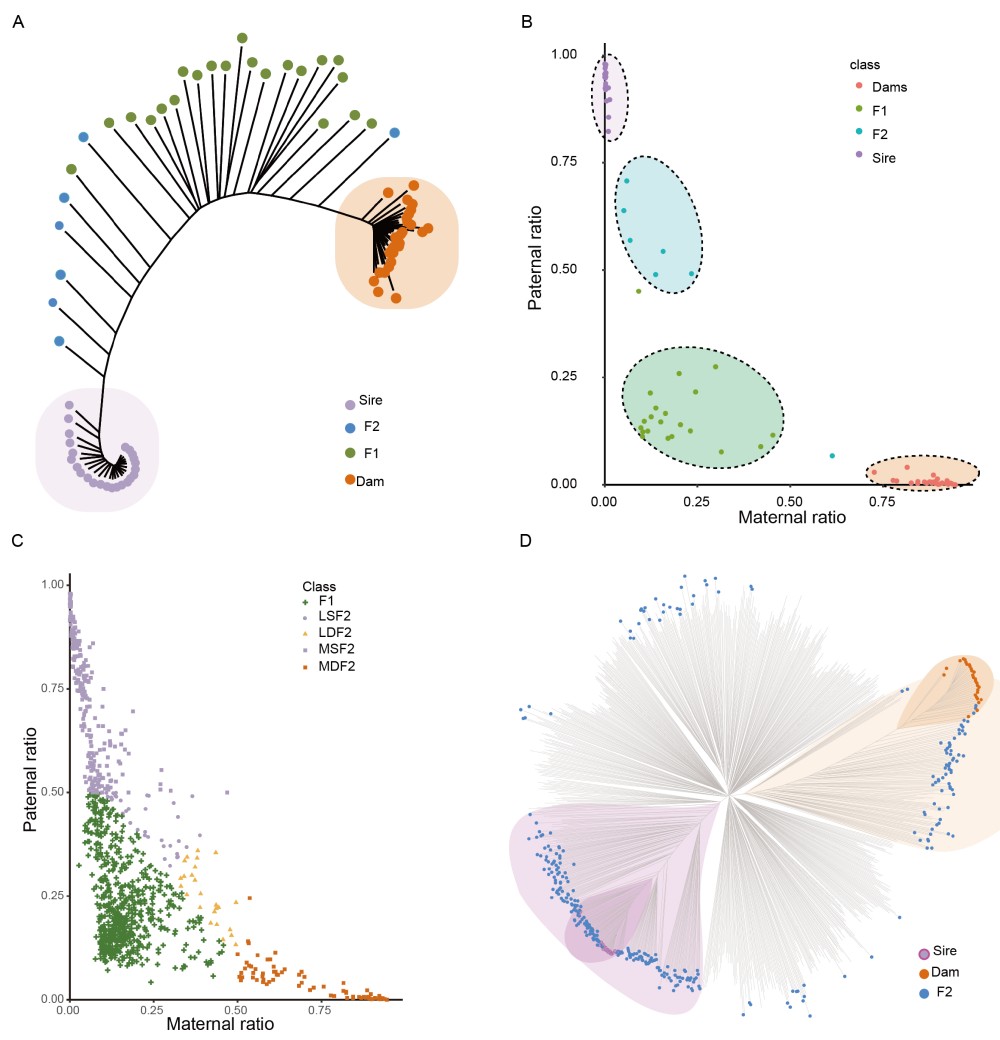

**Figure 4.  Differentiation between the F1 and F2 generations from a group of cattle.**
(A) Phylogenetic tree of 80 samples containing sires, dams, and offspring with known generation information. (B) Scatter diagram of 80 samples according to the paternal and maternal ratios. (C) Scatter diagram of 923 samples according to the paternal and maternal ratios. Different colors represent different subclasses. (D) Phylogenetic tree of all 975 samples containing sires, dams, and offspring.

assigned to the F1 generation but belonged to the F2 generation. It is possible that the F2 individuals were produced from F1 inter-crossing, which influenced the final classification results (Figure 1B). However, considering that the selection of superior individuals with respect to performance traits from sires and dams is the ultimate goal of this crossbreeding system, it is unnecessary to distinguish between these individuals.

The newly developed SNP panel was used to genotype a total of 975 samples. Based on the SNP information, almost all offspring were categorized successfully (Table 3 in GigaDB [29]). Samples were clearly divided into five groups: the F1 generation (heterozygous ratio > 0.5, 615 samples), the F2 generation with strong paternal characteristics (MSF2, heterozygous ratio > 0.5 and paternal ratio > 0.5, 227 samples), the F2 generation with strong maternal characteristics (MDF2, heterozygous ratio > 0.5 and

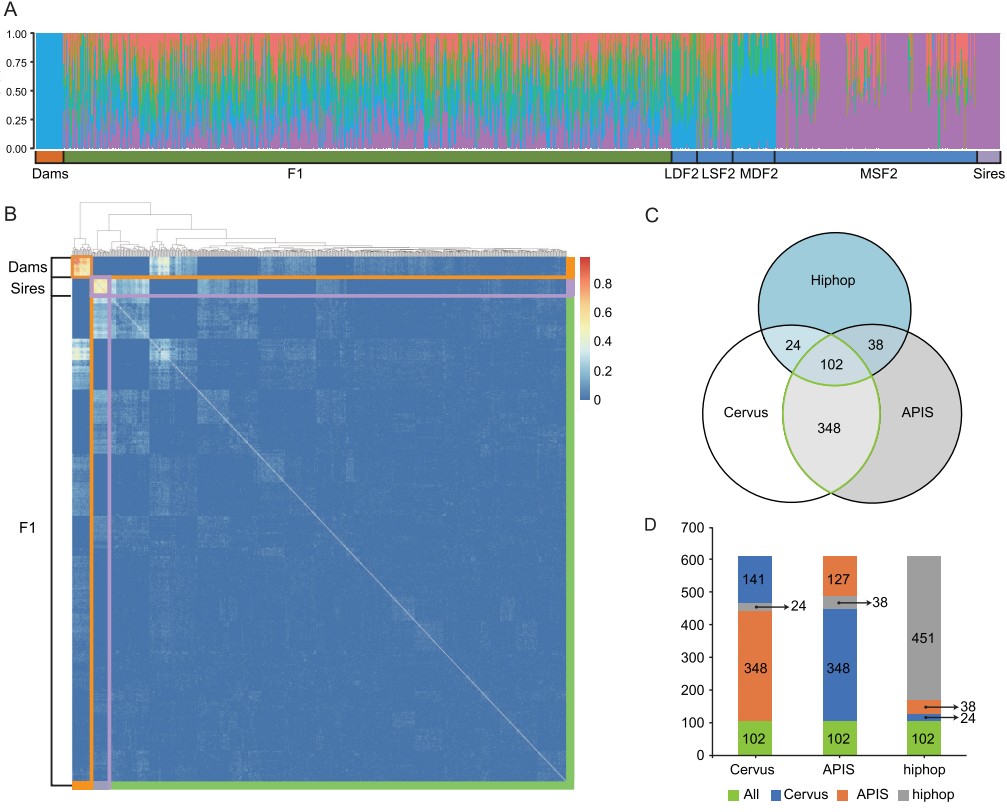

**Figure 5. Parentage assignment of the F1 generation.**
(A) Population structure analysis of 975 samples with $K$ = 5. (B) Heatmap representing the correlations between 615 samples using kinship matrices. (C) Venn diagram indicating parentage assignments obtained using three algorithms. The green oval represents the intersection between Cervus and APIS, defined as the final assignments. (D) Stacked columns representing the composition within each parentage assignment results based on three programs.

maternal ratio > 0.5, 73 samples), the F2 generation with slight paternal characteristics (LSF2, heterozygous ratio > 0.5 and paternal ratio > maternal ratio, 37 samples), and the F2 generation with slight maternal characteristics (LDF2, heterozygous ratio > 0.5, maternal ratio > paternal ratio, 26 samples). The scatter diagram supported the classification of samples into five groups (Figure 4C). Subsequently, we generated a phylogenetic tree based on the information for all samples (Figure 4D). Samples classified as F2 were distributed across the phylogenetic tree, with one group clustered near sires (MSF2) and another near dams (MDF2). The samples located in an intermediate position were considered F1 or F2 with slight parental characteristics (LSF2 and LDF2) (Table 3 in GigaDB [29]).

## Validation of classification results

To confirm the correctness of our classification, we performed a population structure analysis using all SNPs to observe the systematic differences in allele frequencies among subpopulations (Figure 5A). In total, 975 samples were divided into five subpopulations, consistent with the results for the classification of generations.

Gene IBD is a fundamental concept that explains genetically mediated similarities among relatives [31]. The $PI_{HAT}$ value ($\hat{\pi}$, *proportion* IBD) obtained using PLINK provides an

estimate of IBD. This method is based on hidden Markov models and calculates the probability of IBD = 1, 2, or 0 by method-of-moments estimation. The value of PI_HAT is between 0 and 1, and a higher value indicates a closer relationship. We performed an IBD analysis to evaluate relationships among individuals. Usually, pairs with PI_HAT near 1 are considered identical twins, those with values of approximately 0.5 are considered parent/child or parental identical twins, and those with values of around 0.25 are considered half-siblings. We used the homozygous sites from sires and dams to calculate PI_HAT. All pairs with a PI_HAT value > 0.5 were pairs between sires or dams. Most pairs with PI_HAT values around 0.5 (0.3–0.5) were pairs between sires/dams and MSF2/MDF2 (Table 7 in GigaDB [29]). These results were in accordance with our previous assignments to provide a basis for selecting MSF2 for breeding and other F2 for conservation.

Finally, we created kinship matrices for individuals containing sires, dams, and the F1 generation. These were multiplied by two to indicate expected covariances between samples and were used for parentage assignment (Figure 5B and Table 4 in GigaDB [29]). This approach proved effective for distinguishing between heterozygous offspring and candidate homozygous offspring with parental traits.

## Parentage assignment of the F1 generation

For the parentage assignment with the F1 generation, we chose three tools with different underlying algorithms and compared the results. CERVUS is widely used and utilizes a likelihood-based approach [9], APIS uses the observed distribution of Mendelian transmission probabilities, and HIPHOP extends exclusion approaches with SNP markers.

In total, 960 SNPs that met the quality criteria were further filtered to determine loci that could provide sufficient statistical power for parentage assignment. CERVUS requires allele frequency estimation and a simulation before formal assignment. Simulation can be used to examine the feasibility of parentage analysis and calculate critical values of likelihood ratios to determine the confidence of parentage assignments for real data. The following parameters were used for simulation: 30% loci typed, 0.01% genotyping error rate, 10,000 offspring, 50 candidate fathers, 90% of candidate fathers sampled, and the minimum number of typed loci set to 10. The output of strict confidence value of simulation was 95%. Finally, by inputting the simulation and SNP files, the set containing 28 sires and the panel of 960 SNP markers provided 99.5% successful parentage assignment for 615 offspring (Table 5 in GigaDB [28]). Among 612 F1–sire relationships, 552 had confidence levels >80%, including 451 with confidence ≥95%, which indicated a highly significant relationship. The accepted assignment error rate was 0.5 using APIS (Figure 6) [27]. HIPHOP required a known social parent and year of birth for each cattle at the time of parentage assignment [28]. If the individual was the social parent of the brood, then the social parent parameter was set to 1; otherwise, the parameter was set to 0. All cattle were bred in livestock farms, and none were classified as social parents. We then compared the results obtained by the three algorithms and drew a Venn diagram to visually evaluate shared assignments (Figure 5B). The results obtained using CERVUS and APIS overlapped by more than 75% (450 identical assignments), while HIPHOP assignments agreed with those of the other algorithms in <50% of cases (Figure 5C, and Table 5 in GigaDB [29]).

As mentioned previously, HIPHOP relies on exclusion methods. This method identifies incompatibilities between pairs of individuals according to Mendel's laws, and its accuracy depends on the accuracy of the marker data. When the power of the marker set is low, the

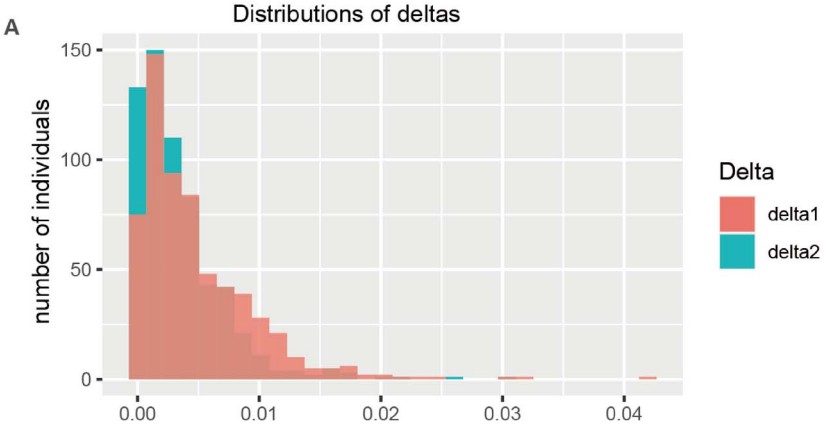

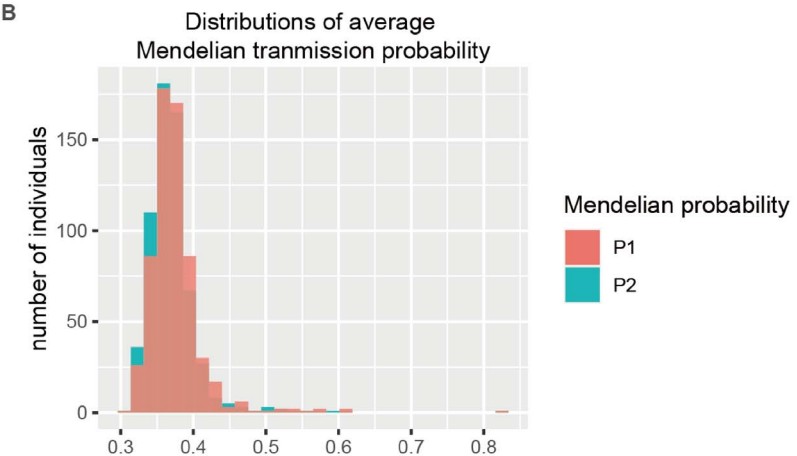

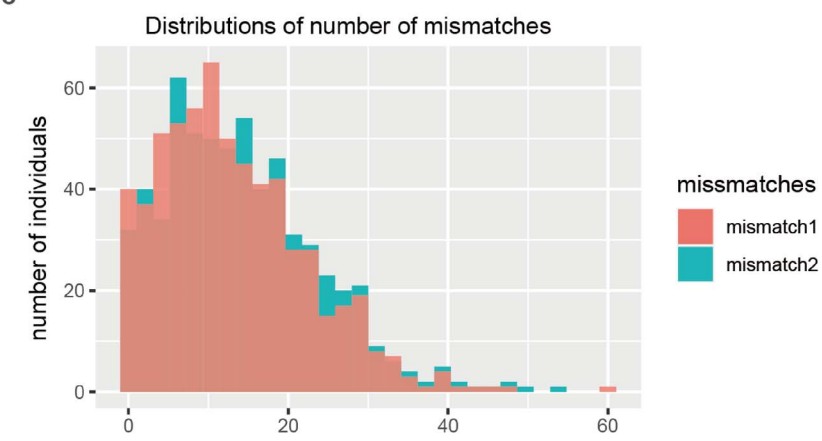

**Figure 6.  Histogram outputs by APIS.**
Delta1: mendel1 - mendel2; Delta2: mendel2 - mendel3. Mendel1, 2, 3: average Mendelian transmission probability of the most likely parent pair (sire1, dam1), second likely parent pair (sire2, dam2), and third likely parent pair (sire3, dam3).

results become unauthentic. However, it is nearly impossible to obtain error-free data as the sample size and number of markers grow. Consequently, the overlapping assignments obtained by CERVUS and APIS were considered the likely parentage assignments with high confidence (Table 6 in GigaDB [29]).

## DISCUSSION

RAD-Seq to genotype SNPs is a practical and cost-effective approach compared to traditional methods such as microarrays. Choosing informative SNPs reduces the computational burden of downstream analyses, and allows us to obtain the desired level of accuracy without generating excessive amounts of data or compromising quality. Overall, RAD-Seq remains an efficient tool for genotyping and has wide applications in population genetics research. Our SNP panel offers a cost-effective tool for livestock management by enabling breeders to make informed decisions on the selection of animals for breeding programs. The detailed genetic information provided by RAD-Seq can lead to improved health, productivity, and overall welfare of the livestock. The SNP markers identified in our study can be used to trace and preserve valuable genetic traits within local breeds. This can facilitate targeted breeding strategies to enhance specific characteristics, such as disease resistance, adaptability, and production efficiency. This SNP panel also can be a valuable tool for conservation efforts by providing a means to monitor and manage the genetic diversity of endangered species. It can also aid in the detection of hybridization events and inform conservation breeding programs. We propose that future studies focus on validating the SNP panel across a broader range of breeds and environments. The SNP panel could be integrated into existing breeding programs, starting with a pilot study to assess its effectiveness and economic benefits. This would help to establish the panel's versatility and robustness in different contexts.

To assign individuals to the F1/F2 generation, a panel of SNPs can be used to determine the number of loci that are identical to the maternal and paternal alleles or are heterozygous. This allows for the calculation of maternal, paternal, and heterozygous ratios for each offspring. Theoretically, the heterozygous ratio for the F1 generation should be around 1.0, while the maximum ratio was 0.83 in our analysis. Furthermore, the F2 generation was divided into four subclasses according to the maternal and paternal ratios. This likely reflects the law of segregation and the law of independent assortment for non-allelic genes on non-homologous chromosomes. In contrast, the law of linkage and crossing-over would lead to deviations from the expected ratio. Unexpectedly, the F2 generation contained the MDF2 and LDF2 subgroups. Considering that the F1 generation originated from the cross between dams and sires and the F2 generation from the backcrosses between the F1 generation and sires theoretically, this observation can likely be explained by crossing within the F1 generation, since the proportions of MDF2 and LDF2 in the whole F2 generation were low. In addition, during the SNP selection stage, we allowed five different alleles, which could be the result of sequencing error. This is another reason why the heterozygous ratio of the F1 generation was < 1.

Although several methods for parentage analysis have been developed, accurate parentage assignment critically depends on the establishment of a molecular marker panel. Accurate pedigree information has been determined in a Mexican-registered Holstein population [32] and in Chinese Simmental cattle based on a high-density SNP array [33]. Most parentage assignments rely on a microsatellite chip or SNP chip. In taxa, without

accurate SNP chips for parentage assignment, it is difficult to achieve this goal. RAD-seq and WGS provide abundant variant information, which enables parentage and population analyses at the same time. Of note, the SNP panel used for parentage assignment should be optimized based on several factors, including linkage disequilibrium, minor allele frequencies, deviation from Hardy–Weinberg equilibrium, genotyping errors, and the frequency of null alleles. In this study, we achieved parentage assignment by extracting the intersection between CERVUS and APIS results. However, 165 offspring could not be assigned to accurate sires. This could probably be attributed to an inherent defect of RAD-seq, resulting in a high frequency of null alleles. Null alleles cannot provide accurate predictions, thus weakening the confidence of the prediction results.

Another feasible approach to achieving parentage assignment is based on the genotypes of sex chromosomes and mitochondrial chromosomes. Typically, the X and Y chromosomes do not undergo standard recombination in males, and mitochondrial chromosomes are inherited directly from the mother [31, 34, 35]. Therefore, alleles on the sex chromosome and mitochondria are transferred to offspring directly. The Y chromosome is difficult to assemble because it contains many ampliconic and palindromic sequences [36]. Also, SNPs and InDels were sparsely distributed on the X chromosome (Figures 2 and 3) except for the tip region. In conclusion, genotyping by RAD-seq is an efficient method for the classification of generations and parentage assignment. Individuals from different generations identified through the analysis can be used to accelerate the subsequent breeding process and breeding conservation.

## CONCLUSION

Crossbreeding is a widely used and effective tool to increase genetic diversity within a breed. However, successful crossbreeding relies on accurate marking and recording of newborn calves, which is laborious and increases in difficulty as the calf population grows. Traditional SNP genotyping may be more suitable for smaller projects or when specific SNPs are the focus of the research. RAD sequencing is generally considered a more cost-effective and scalable solution for large-scale SNP genotyping projects due to its lower per-sample costs and ability to multiplex many samples simultaneously. The choice between the two methods should be based on the specific requirements of the project, including the number of samples, the number of SNPs to be genotyped, and the available resources. In this study, we performed RAD sequencing and identified the F1 and F2 generation from Angus cattle and Xiangxi yellow cattle crosses according to Mendelian laws of inheritance and selected an SNP panel with high confidence for kinship analysis and pedigree reconstruction. The F1 generation and MSF2 can be used for breed selection, and the LDF2 and MDF2 generation can be maintained for breed conservation. To the best of our knowledge, this is the first application of a RAD-seq-based approach for simultaneous generation classification and parentage assignment. The combination of the efficiency of RNA-seq and advances in kinship analysis is expected to improve breed management, local resource utilization, and conservation.

## DATA AVAILABILITY

The raw sequencing reads are deposited at NCBI under PRJNA1063367, the SNP file is deposited in figshare [37], and additional data is in GigaDB [29].

## ABBREVIATIONS

IBD, identity-by-descent; InDel, insertion-deletion; MGL, maternal genotype locus; PGL, paternal genotype locus; RAD-seq, restriction-site-associated DNA sequencing; SNP, single nucleotide polymorphism; VCF, Variant Call Format; WGS, whole-genome resequencing.

## DECLARATIONS

### Ethics approval and consent to participate

All of the experimental procedures involving animals were conducted following local guidelines for animal experimentation and were approved by the institutional review board of BGI (approval number BGI-IRB E22013).

### Competing interests

The authors declare that they have no competing interests.

### Authors' contributions

JJ, YXu and JH conceived the study. YXu, JH, MX, YW and YXie collected the samples, conducted experiments, BW, WW and JJ performed bioinformatics analysis. WW, JJ, YXu and JH wrote the manuscript. All authors have read and approved the final manuscript.

### Funding

No external funding was used in this work.

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
