## [Editor Report]

Editor’s AssessmentRAD-Seq (Restriction-site-associated DNA sequencing) is a cost-effective method for single nucleotide polymorphism (SNP) discovery and genotyping. In this study the authors performed a kinship analysis and pedigree reconstruction for two different cattle breeds (Angus and Xiangxi yellow cattle). A total of 975 cattle, including 923 offspring with 24 known sires and 28 known dams, were sampled and subjected to SNP discovery and genotyping using RAD-Seq. Producing a SNP panel with 7305 SNPs capturing the maximum difference between paternal and maternal genome information, and being able to distinguish between the F1 and F2 generation with 90% accuracy. Peer review helped highlight better the practical applications of this work. The combination of the efficiency of RNA-seq and advances in kinship analysis here can helpfully help improve breed management, local resource utilization, and conservation of livestock.Editor’s AssessmentRAD-Seq (Restriction-site-associated DNA sequencing) is a cost-effective method for single nucleotide polymorphism (SNP) discovery and genotyping. In this study the authors performed a kinship analysis and pedigree reconstruction for two different cattle breeds (Angus and Xiangxi yellow cattle). A total of 975 cattle, including 923 offspring with 24 known sires and 28 known dams, were sampled and subjected to SNP discovery and genotyping using RAD-Seq. Producing a SNP panel with 7305 SNPs capturing the maximum difference between paternal and maternal genome information, and being able to distinguish between the F1 and F2 generation with 90% accuracy. Peer review helped highlight better the practical applications of this work. The combination of the efficiency of RNA-seq and advances in kinship analysis here can helpfully help improve breed management, local resource utilization, and conservation of livestock.

---

## [Reviewer Report]

Reviewer name and names of any other individual's who aided in reviewer Liyun WanDo you understand and agree to our policy of having open and named reviews, and having your review included with the published papers. (If no, please inform the editor that you cannot review this manuscript.)YesIs the language of sufficient quality?YesPlease add additional comments on language quality to clarify if needed
Are all data available and do they match the descriptions in the paper? YesAdditional CommentsAre the data and metadata consistent with relevant minimum information or reporting standards? See GigaDB checklists for examples <a href="http://gigadb.org/site/guide" target="_blank">http://gigadb.org/site/guide</a>YesAdditional CommentsIs the data acquisition clear, complete and methodologically sound?YesAdditional CommentsIs there sufficient detail in the methods and data-processing steps to allow reproduction?YesAdditional CommentsThe detailed parameters for the SNP and InDel calling should be described to allow reproduction.Is there sufficient data validation and statistical analyses of data quality? YesAdditional CommentsIs the validation suitable for this type of data?YesAdditional CommentsIs there sufficient information for others to reuse this dataset or integrate it with other data?YesAdditional CommentsAny Additional Overall Comments to the AuthorThis research provides valuable insights into the use of RAD-Seq to kinship analysis and pedigree reconstruction, which is useful for breeding and animal conservation purposes. Overall, the study is well-conducted and the findings are relevant. However, there are a few aspects that require attention before the manuscript can be considered for publication. Please address the following points: 1. Provide practical applications: Highlight the practical applications of your research in livestock management, breed improvement, local resource utilization, and conservation. Discuss how the low-cost and efficient SNP panel can contribute to these areas and provide suggestions for further research or implementation. 2. Language and clarity: Review the manuscript for clarity, grammar, and sentence structure. Ensure that all key terms and concepts are defined and explained to facilitate understanding for a broad readership. Once these revisions have been made, I believe the manuscript will be much stronger and suitable for publication. RecommendationMinor Revision

---

## [Reviewer Report]

Upload additional filesDRR-202401-07-R01/stage_files/DRR-202401-07/Review MS/giga-coments.docxReviewer name and names of any other individual's who aided in reviewer Mohammad Bagher ZandiDo you understand and agree to our policy of having open and named reviews, and having your review included with the published papers. (If no, please inform the editor that you cannot review this manuscript.)YesIs the language of sufficient quality?YesPlease add additional comments on language quality to clarify if needed
It was greatAre all data available and do they match the descriptions in the paper? YesAdditional CommentsThe raw sequencing reads were deposited but it would be better to share the the SNPs data as well.Are the data and metadata consistent with relevant minimum information or reporting standards? See GigaDB checklists for examples <a href="http://gigadb.org/site/guide" target="_blank">http://gigadb.org/site/guide</a>YesAdditional CommentsIs the data acquisition clear, complete and methodologically sound?NoAdditional CommentsSNPs detection and SNPs selection for assignment test is not clear.Is there sufficient detail in the methods and data-processing steps to allow reproduction?NoAdditional CommentsIn some cases, the materials and methods section is vague It is better to correct them It is mentioned in the attached manuscript textIs there sufficient data validation and statistical analyses of data quality? YesAdditional CommentsIs the validation suitable for this type of data?YesAdditional CommentsIs there sufficient information for others to reuse this dataset or integrate it with other data?YesAdditional CommentsAny Additional Overall Comments to the AuthorWell done research, but the manuscript need some correction as commented on the attached file.RecommendationMinor Revision